# Targeting the Class A Carbapenemase GES-5 via Virtual Screening

**DOI:** 10.3390/biom10020304

**Published:** 2020-02-14

**Authors:** Raphael Klein, Laura Cendron, Martina Montanari, Pierangelo Bellio, Giuseppe Celenza, Lorenzo Maso, Donatella Tondi, Ruth Brenk

**Affiliations:** 1Institute of Pharmacy and Biochemistry, Johannes Gutenberg University, 55122 Mainz, Germany; 2Department of Biology, University of Padua, Viale G. Colombo 3, 35121 Padua, Italy; 3Department of Life Sciences, University of Modena and Reggio Emilia, Via Campi 103, 41125 Modena, Italy; 4Department of Biotechnological and Applied Clinical Sciences, University of L’Aquila, via Vetoio 1, 67100 L’Aquila, Italy; 5Department of Biomedicine, University of Bergen, Jonas Lies Vei 91, 5020 Bergen, Norway

**Keywords:** antibiotic resistance, GES-5, Guyana extended-spectrum-β-lactamase, carbapenemase, virtual screening, docking, noncovalent inhibition

## Abstract

The worldwide spread of β-lactamases able to hydrolyze last resort carbapenems contributes to the antibiotic resistance problem and menaces the successful antimicrobial treatment of clinically relevant pathogens. Class A carbapenemases include members of the KPC and GES families. While drugs against KPC-type carbapenemases have recently been approved, for GES-type enzymes, no inhibitors have yet been introduced in therapy. Thus, GES carbapenemases represent important drug targets. Here, we present an in silico screening against the most prevalent GES carbapenemase, GES-5, using a lead-like compound library of commercially available compounds. The most promising candidates were selected for in vitro validation in biochemical assays against recombinant GES-5 leading to four derivatives active as high micromolar competitive inhibitors. For the best inhibitors, the ability to inhibit KPC-2 was also evaluated. The discovered inhibitors constitute promising starting points for hit to lead optimization.

## 1. Introduction

The spread of multidrug resistant (MDR) Gram-negative bacteria is a major public health threat [1]. While lately the attention to antimicrobial resistance and to antibiotic drug discovery has grown considerably, with novel bacterial targets being investigated [2,3,4] the necessity for new antimicrobials remains urgent. At present, β-lactams represent, worldwide, the most used antibiotics against infections caused by Gram-negative bacteria, constituting 40% of all prescribed antibiotics. Therefore, the continuous emergence of β-lactamases (BLs) such as carbapenemases conferring resistance to almost all available β-lactam antibiotics, poses a serious health threat to the public.

Class A carbapenemases include members of the KPC and GES families. KPC- and GES-type enzymes are both plasmid-mediated BLs, thus disseminating horizontally. While KPC-2 inhibitors are at present available for therapy, no compounds active against GES-type BLs are yet available. GES-type BLs, the target of our study, represent a unique example of this family of hydrolytic enzymes. Isolated in the late twentieth century, GES-type BLs were initially classified as extended-spectrum BLs (ESBLs). More recently, because of the documented carbapenemase activity, as well as a structural characteristic shared by other serine carbapenemases (i.e., the disulfide bond between Cys69 and Cys239), GES enzymes have been included among class A carbapenemases, along with KPC-type BLs [5]. GES-1, described for the first time 20 years ago, is structurally similar to other class A ESBLs and confers resistance to extended-spectrum cephalosporins. However, GES-1 has only a low ability to hydrolyze carbapenems, and it is actually inhibited by the carbapenem antibiotic imipenem [6]. Since 2000, thirty-seven variants have been identified and characterized throughout the world, accounting for the evolution of the GES-enzyme spectrum of action from ESBLs to carbapenemases and justifying their inclusion in the group of class A carbapenemases [5,7,8]. At present, GES-type BLs are frequently reported in *Pseudomonas aeruginosa*, *Acinetobacter baumannii*, *Escherichia coli*, and *Klebsiella pneumoniae* [9,10]. The peculiarity GES enzymes share is their ability to extend their hydrolytic spectrum to carbapenems by single point mutations [11]. Among the detected variants, a single mutation at position 170 (from Gly170 to Asn/Ser170) is recognized as critical for their ability to hydrolyse last resort β-lactams. As a matter of fact, this mutation is correlated with higher hydrolytic activity against imipenem. Importantly, the *bla*_GES_ genes have been described as gene cassettes associated with integrons on plasmids, explaining their horizontal, thus rapid dissemination [8]. Among GES-type BLs, GES-5 exhibits higher carbapenem-hydrolyzing activities and nosocomial infections caused by GES-5-producing strains that often respond only to amikacin and colistin, representing a growing concern [10].

Thirteen crystal structures of GES-type BLs are now available, making it an attractive target for the structure-based design of novel molecules able to counteract antibiotic resistance mediated by these enzymes [12,13,14,15,16,17,18]. However, compared to other carbapenemases, the number of available GES-type BL crystal structures is still quite limited, reflecting the low level of attention so far dedicated to the GES family. For GES-5, the target of our study, five crystal structures of the wild-type or mutant enzyme have been deposited in the Protein Data Bank (PDB) [12,13,14]. Two of them are binary complexes with either the substrate imipenem or a boronic acid inhibitor. A characteristic feature of BL binding sites is the oxyanion hole, which is typically addressed by negatively charged groups. In the case of GES-5, the oxyanion hole is formed by the amino acids Ser125, Thr211, Thr230 and Thr232 (Figure 1).

Very few GES-5 inhibitors are known. We have recently reported on boronic acids as covalent inhibitors for class A and C BLs, including GES-5 [14,19]. Further, we have conducted the first in silico modeling against this clinically relevant enzyme, leading to the identification of six novel inhibitors [20]. These compounds have *K*_i_ values ranging from 0.16 to 0.50 mM, which translates to ligand efficiencies (LE) of 0.16 to 0.26 kcal/mol/non-hydrogen atom. The compounds are rather hydrophobic, leading to solubility issues and making them less ideal starting points for hit optimization.

In the work presented here, we set out to identify more soluble inhibitors of GES-5 carbapenemase with a higher ligand efficiency using a structure-based virtual screening approach against an in-house determined crystallographic structure. Forty-four compounds were selected for experimental testing against GES-5. These compounds were mostly fragment-like and predicted to bind into the oxyanion hole. The most potent compounds were also tested towards KPC-2. Four fragments were identified with *K*_i_ values for GES-5 and KPC-2 in the high micromolar to the low millimolar range. Two of them had LEs > 0.29 kcal/mol/nonhydrogen atom against GES-5. They constitute promising starting points for hit to lead optimization.

## 2. Materials and Methods

### 2.1. Pharmacophore Hypothesis

A pharmacophore for GES-5 inhibitors was derived in an analogous way as done previously for KPC-2 [21]. The structure comparison of GES-5 and KPC-2 was performed using the Molecular Operating Environment (MOE; Chemical Computing Group, Montreal, QC, Canada) and PyMOL (Schrödinger, LLC, New York, NY, USA). The published structures of GES-5 (PDB code 4GNU and 4H8R) and our in-house structure were aligned with the structure of KPC-2 (PDB code 3RXW) to identify key interactions. The final pharmacophore was based on the apo-crystal structure of GES-5, which we have solved for this project (PDB code 6TS9), and the ligand 0JB of CTX-M-9 β-lactamase (PDB code 4DE0). It contained the same features as our previous pharmacophore for KPC-2, namely a hydrogen-bond acceptor feature for interactions with Ser125, Thr230 and Thr232, a hydrophobic π-stacking feature with Trp99 and a hydrogen bond acceptor feature for interaction with Asn127 (Figure 2a). The interactions to Thr230 and Thr232 were set as mandatory for filtering the obtained docking hit list.

### 2.2. Virtual Screening

The in-house crystal structure of GES-5 (PDB code 6TS9) was used as a receptor for docking. Flexibility of the binding site residue Trp99, key for the carbapenemase hydrolytic spectrum, was assumed. Therefore, two different docking setups were prepared to take into account the flexibility of Trp99 previously observed for KPC-2 (Figure 2b). For that purpose, the “Protein Builder” function of MOE was used to introduce a rotamer of Trp99 as found in KPC-2. Using the ‘protonate 3D’ tool of MOE, polar hydrogen atoms were added to the receptor, and their positions were energy minimized while keeping the protein atoms rigid [22]. Subsequently, all water molecules were deleted. To define the matching points for docking, the structure was aligned with the crystal structure of *E. coli* CTX-M-9 (PDB code 4DE0), and the benzimidazole atoms of the ligand were transferred to the GES-5 structure. Information about excluded volumes, van der Waals potential, electrostatic potential and solvent occlusion maps were stored in Grids and calculated as described earlier [23,24].

DOCK3.6 was used for docking the compounds into the two setups of the GES-5 binding site [24,25,26]. The two receptor setups were validated with a training set consisting of the ligands GF4 (PDB code 3G2Y), F13 (3G35), 0JB (4DE0) 0J6 (4DE1) and DN3 (4DE2). The best results in terms of binding mode prediction were obtained when the magnitudes of the partial charges of some atoms of Thr232 were adjusted without changing the net residue charge (an increase of partial charges of HN and HOG by 0.4 and decrease of partial charges of O and OG1 by –0.4). Changing the dipole moments of these atoms in such a way favored the formation of a hydrogen bond between the ligands and this residue. Next, docking was carried out as described previously against the same set of mostly fragment-like commercially available compounds which fulfilled the following criteria: between 10 and 25 heavy atoms, between one and six hydrogen-bond acceptors, between one and three hydrogen-bond donors, a clogP between −3 and 3, a limited complexity with less than 7 rotatable bonds and between one and three-ring systems [21]. Hits for which their predicted binding mode passed the pharmacophore filter described above were ranked by their calculated LE and inspected by eye.

### 2.3. Cloning, Expression and Purification of Recombinant bla_ges-5_

For protein expression, competent *E. coli* BL21(DE3) cells (PROMEGA, Madison, WI, USA) were transformed with recombinant plasmid *bla*_GES-5_/pet24a(+) as reported [20]. Protein purification was performed using an Äkta Prime plus chromatograph system (GE Healthcare Life Sciences, Freiburg, Germany). GES-5 (Uniprot ID Q09HD0) was conveniently purified in a single step using a Macro-Prep High Q resin (Bio-Rad, Hercules, CA, USA.). For the obtained protein, the *K*_M_ values for the two reporter substrates, CENTA and nitrocefin were determined and found to be 524 μM and 208 μM, respectively [27].

### 2.4. Crystallization and Structure Determination

Purified GES-5 carbapenemase (Uniprot ID Q09HD0) was gel filtered in a buffer (25 mM Tris, 100 mM NaCl, pH 7.5) to avoid the presence of HEPES buffer for crystallization. Crystallization, data collection and structure determination were performed as already reported [14]. The determined structure contained two molecules per asymmetric unit and was refined to R-free/R-work values of 0.18/0.20 with a resolution of 1.55 Å (Appendix A). In the active site, an electron density blob close to Ser125 and Thr230 was interpreted as dimethyl sulfoxide, while 4 Br atoms were detected in a peripheral region, mediating crystal contacts. The PDB file was deposited with PDB code 6TS9.

### 2.5. In Vitro Validation

The hydrolysis reaction that constitutes GES-5 activity was evaluated using the β-lactam substrate nitrocefin (200 μM, *K*_M_ 208 μM) in a reaction buffer that consisted of 50 mM phosphate buffer and 50 mM KCl at pH 7.4 and 25 °C, with 0.01% *v/v* Triton X-100 to avoid compound aggregation and promiscuous inhibition [28]. Reactions were monitored using a Jasco V730 spectrophotometer. The appearance of the nitrocefin hydrolysis product was monitored at 485 nm. The compounds were purchased from Molport (https://www.molport.com/shop/index) and tested without further purification (purity ≥ 90%). In order to perform spectrophotometric tests, compounds were dissolved in dimethyl sulfoxide (DMSO) to a concentration of 25 mM and stored at −20 °C. Almost all compounds showed a good solubility at 25 mM concentration. Exceptions were compounds 13 and 42 that were solubilized at 12.5 mM, and compounds 11 and 38 that were not tested because of solubility issues (Appendix A). The highest concentration at which the compounds were tested depended on their solubility and, in the best cases, went up to 3 mM. All the experiments were performed in duplicate, and the errors never exceeded 5%. The reaction was typically initiated by adding GES-5 (2 nM final concentration), previously dialyzed in the reaction buffer. For the best inhibitors, IC_50_ were determined, and the *K*_i_ for the most active compounds were calculated using the Cheng–Prusoff equation [*K*_i_ = IC_50_/(1 + [S]/*K*_M_)] as per competitive inhibition [29]. For KPC-2 experiments, the protein was obtained, and assays were conducted as previously reported [30].

## 3. Results

The binding site of GES-5 carbapenemase was analyzed in order to generate a 3D-pharmacophore hypothesis. At the outset of the study, five structures of GES-5 were available in the PDB (PDB codes: 4GNU, 4H8R, 5F82, 5F83, 6Q35) [12,13,14]. In addition, we had determined an in-house structure (PDB code 6TS9, Appendix A). The alignment of these structures revealed that the binding site was rigid, with root-mean-square deviation (rmsd) values of the active site residues ranging from 0.094 to 0.274 Å (Figure 3a). The structure of GES-5 in the covalent complex with imipenem revealed hydrogen-bond interactions to Thr230, Asn127 and the backbone NH groups of Thr232 and Ser64 (Figure 1). The binding sites of GES-5 and KPC-2 were structurally very similar (Cα-rmsd between 0.433 and 0.462 Å), and the residues interacting with imipenem in GES-5 were conserved in KPC-2 (Figure 2b). Thus, also structures of small molecules in complex with KPC-2 were considered to include more diverse ligands (PDB codes 3RXW, 3RXX, 4ZBE). The interactions of KPC-2 ligands were recently analyzed [21]. It was revealed that ligands cocrystalized with KPC-2 form hydrogen bonds to the amino acids corresponding to Asn127, Ser125 and Thr232 in GES-5. These residues were part of the oxyanion hole and were therefore considered to be also important for ligands binding to GES-5 (Figure 1). The major difference between GES-5 and KPC-2 was the substitution of Ser170 and Ala243 in KPC-2 (KPC-2 numbering) with Asp166 and Arg238 in GES-5 (GES-5 numbering, Figure 3b). Asp166 did not form any contacts with the bound ligands, whereas Arg238 was part of the hydrogen bond interaction pattern of imipenem (Figure 1). As both binding sites were very similar, we hypothesized that the pharmacophore previously derived for KPC-2 ligands with hydrogen bond interactions to Ser125, Thr232, Thr230, Asn127 and a hydrophobic interaction feature with Trp99 was also applicable to GES-5 (Figure 2a).

As the receptor for docking, the in-house crystal structure of apo GES-5 was used as the target (resolution 1.55 Å). In KPC-2, flexibility of an active site tryptophan was observed (corresponding to Trp99 in GES-5, Figure 3b). We assumed that this residue could also adopt different conformations in GES-5. Therefore, two different structures were used for docking, one containing the rotamer found in GES-5 and one containing the rotamer found in KPC-2.

A hierarchical approach was adopted for virtual screening. We started with the pharmacophore-filtered library of lead-like compounds from our previous quest for KPC-2 inhibitors [21]. The resulting 44,658 compounds were subsequently docked into the two receptor conformations of GES-5. 29,558 compounds could be docked into the binding site of GES-5 as found in the crystal structure, and 29,730 into the binding site of GES-5 with the KPC-2 like Trp99 rotamer. Filtering the docking hit list with the derived GES-5 pharmacophore while keeping the ligand positions rigid, resulted in 10,297 compounds for the GES-5 receptor and 8,898 compounds for the GES-5 receptor with the KPC-2-like Trp99 rotamer. The obtained compounds were clustered according to the functional groups placed in the hydrophilic pocket (tetrazoles, carboxylates or others) and inspected by eye. Finally, 44 compounds were selected for hit validation. Twenty-three of them were placed in the GES-5 crystal structure, 20 in the receptor with the KPC-2-like Trp99 rotamer, and one was predicted to bind to both receptor conformations (Appendix A).

The selected hits were validated in biochemical assays against recombinant GES-5. As most of the compounds were fragment-like, they were tested at 1 mM concentration in the assay. For 37 of the selected 44 compounds, percentage inhibition values could be determined (Figure 4). The remaining seven compounds were not soluble under assay conditions. Nine compounds showed more than 30% inhibition with compound 43 being the most active one, with 41% inhibition (Appendix A; Figure 4a). A subset of the compounds was also tested against KPC-2 to evaluate their ability to inhibit other class A carbapenemases (Figure 4b). This subset included all compounds that showed at least 28% inhibition at 1 mM for GES-5 and less potent compounds that appeared to be very soluble under the assay conditions. For five of the nine of the most potent GES-5 inhibitors, KPC-2 was also inhibited by at least 30% (6, 8, 12, 13 and 24). The compounds 23 and 28 were the strongest KPC-2 inhibitors with ≥40% inhibition, while inhibition of GES-5 was <30%. For compounds 8, 12, 13 and 43, IC_50_ values were determined and their *K*_i_ values were calculated (Table 1 and Appendix A) [29]. As the substrate concentrations in the assays were higher than their *K*_M_ values, the *K*_i_ values were lower than the corresponding IC_50_ values and also more favorable than what might have been expected, based on the percentage inhibition values. The *K*_i_ values for GES-5 were found to be in the high micromolar to the low millimolar range, translating to LEs between 0.22 and 0.32 kcal/mol/nonhydrogen atom. Except for compound 12, KPC-2 was more strongly inhibited by these compounds than GES-5, with *K*_i_ values ranging from 0.28 mM to 1.38 mM.

As required by the derived pharmacophore model, in the predicted binding modes (Table 1) the negatively charged functional groups of the four best compounds were placed in the oxyanion hole, mimicking the interactions of the carboxylate group of the β-lactam ring of imipenem (Figure 1 and Figure 2a). Compounds 12 and 43 were predicted to bind to a binding site conformation in which Trp99 was rotated towards the ligands resulting in π-π interactions with the bound compounds.

## 4. Discussion

Virtual screening for GES-5 carbapenemase inhibitors resulted in a number of weakly inhibiting compounds (Figure 4, Appendix A). For four of the most potent compounds, inhibition values were determined (Table 1). These compounds are chemically diverse, including a tetrazole-derivate (8) two aromatic acids (12 and 43) and an aliphatic acid (13). Two of them (13 and 43) had LEs ≥0.29 kcal/mol/nonhydrogen atom, making them attractive starting points for hit to lead optimization.

The binding sites of GES-5 and KPC-2 were very similar, especially at the level of the targeted oxyanion hole (Figure 3). Therefore, it is not surprising that the hits obtained against GES-5 also showed activity against KPC-2 (Figure 4, Table 1 and Appendix A). The compound with the highest LE found in this study (43), had already been reported as a hit in our previous virtual screening against KPC-2 [16]. The benzothiazole derivate of this compound showed a 6-fold improved activity for KPC-2, suggesting that this derivative could also be more active against GES-5. The good LE for the top hits for both GES-5 and KPC-2 opens up the possibility to develop dual inhibitors for these enzymes. However, moving away from the oxyanion hole, the two binding sites become more different (Figure 3b). Thus, subsequent optimization programs will prove if potent dual inhibition is indeed achievable.

Compared to our previous virtual screening, the hits obtained here are slightly less potent, but have an improved ligand efficiency, and, most importantly, better solubility. In the previous study, the top hit in terms of affinity had a *K*_i_ of 0.16 mM and an LE of 0.22 kcal/mol/nonhydrogen atom, and the top hit in terms of LE had a *K*_i_ of 0.50 mM and a LE of 0.26 kcal/mol/nonhydrogen atom [20]. Here, the top hit has a *K*_i_ of 0.66 mM and a LE of 0.32 kcal/mol/non-hydrogen atom. Interestingly, in contrast to our previous study, nearly all compounds short-listed for testing were soluble under assay conditions which allowed to assess nearly all selected compounds in the in vitro tests. This is probably due to the more aggressive filtering of the screening compounds in terms of logP and molecular size, thus leading to candidates with improved solubility. Together with their good LEs, this renders them suitable for hit-to-lead optimization.

As expected in a fragment-based design approach, the discovered hits are not very potent [31]. In addition, the retrieved compounds are noncovalent inhibitors, while many potent BL inhibitors bind covalently to their target, further contributing to the unfavorable potency. Due to their small size, the hits are unable to fill the binding site properly. In turn, this suggests their affinity can be improved by growing the compounds to allow them to form additional interactions with the active site.

## 5. Conclusions

The applied methodology allowed the identification of noncovalent GES-5 carbapenemase inhibitors. Using a library of commercially available compounds, 44 candidates were selected for in vitro validation in spectrophotometric assays against isolated GES-5. Three diverse hits were found to be high micromolar inhibitors for this clinically relevant carbapenemase, for which only very few inhibitors are known, despite its increasing and worldwide spread. Characterized by low affinity, but good LEs, the identified compounds represent novel chemotypes for the development of GES-5 inhibitors with improved potency. They have been directed to hit-to-lead optimization and X-ray crystallographic studies to elucidate their binding modes and to increase their chemical complexity and affinity towards GES-5 carbapenemase.

## Figures and Tables

**Figure 1 biomolecules-10-00304-f001:**
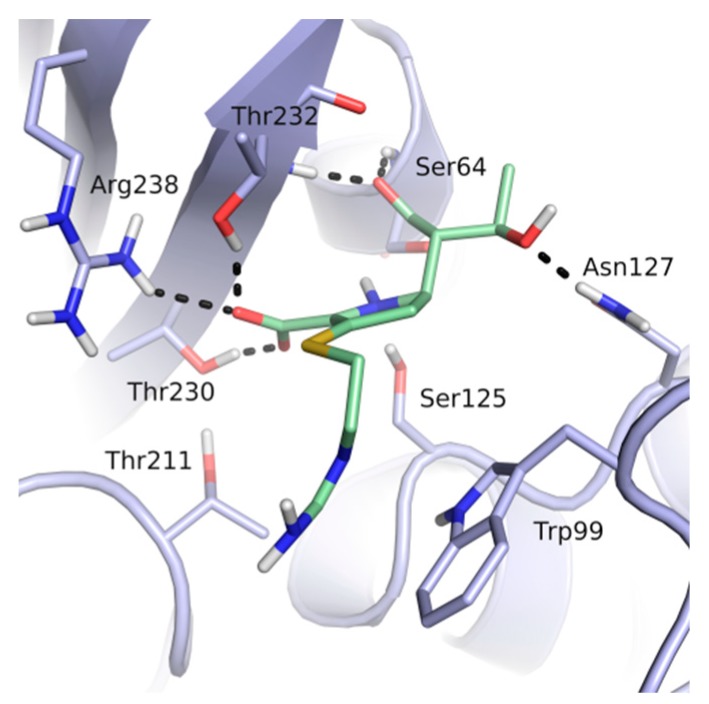
The binding site of GES-5 (PDB code 4H8R) with ligand imipenem (green). Hydrogen bonds are indicated as black dots. The carboxylate group of imipenem is oriented in the oxyanion hole formed by Ser125, Thr211, Thr230 and Thr232.

**Figure 2 biomolecules-10-00304-f002:**
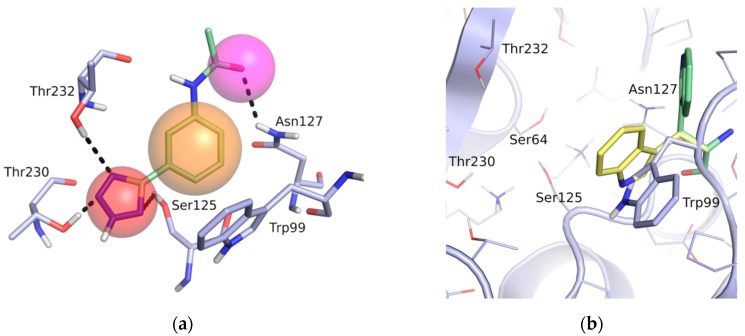
(**a**) Pharmacophore model for GES-5 ligands. Pharmacophore features (red: hydrogen-bond acceptor, orange: hydrophobic interaction feature, purple: hydrogen-bond donor) in the binding site of GES-5 superimposed with a fragment of the ligand 0J6 bound to CTX-M-9 β-lactamase (PDB code 4DE1). (**b**) Conformation of Trp99 found in GES-5 (PDB code 6TS9, blue) and KPC-2 (PDB code 3RXW, yellow: conformation A, green: conformation B). Conformation A of KPC-2 and the conformation found in GES-5 were used for virtual screening.

**Figure 3 biomolecules-10-00304-f003:**
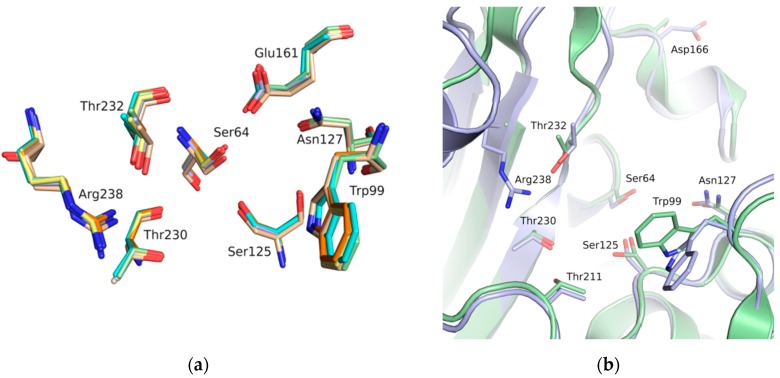
(**a**) Alignment of binding site residues of available GES-5 structures (PDB 6TS9, green; PDB code 6Q35, beige; 4GNU, blue; 4H8R, yellow; 5F82, cyan; 5F83, orange). (**b**) Alignment of the binding sites of GES-5 (blue, PDB code 6TS9) and KPC-2 (green, PDB code 3RXW) The residue numbering is for GES-5.

**Figure 4 biomolecules-10-00304-f004:**
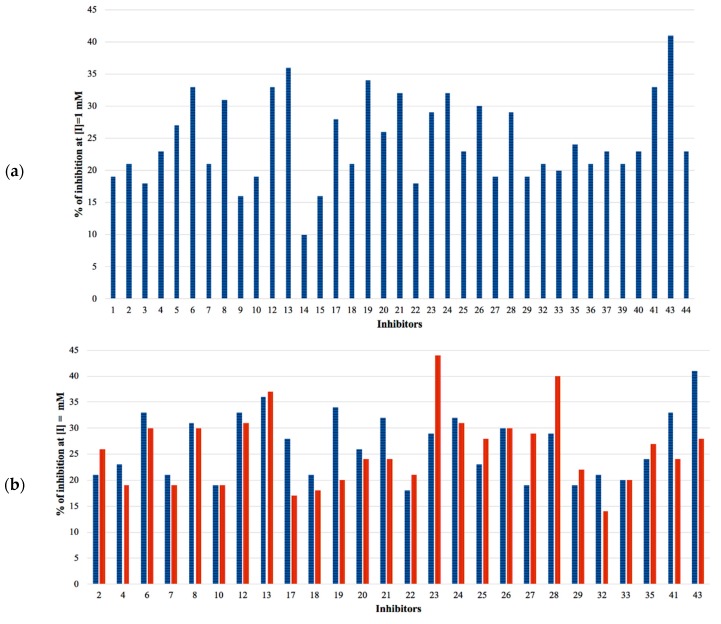
(**a**) Percentage inhibition against GES-5 at 1 mM inhibitor concentration. (**b**) Percentage inhibition at 1 mM inhibitor concentration against GES-5 (blue) and KPC-2 (red).

**Table 1 biomolecules-10-00304-t001:** Inhibition constants, LEs and predicted binding modes of the most potent compounds. The structure with Trp99 in the conformation found in GES-5 is colored in beige; the structure with KPC-2 conformation of Trp99 in blue. The compounds are colored in green; hydrogen bond interactions are indicated as black dots.

Code	*K*_i_ [mM] vs. GES-5 (LE) [a]	*K*_i_ [mM] vs. KPC-2 (LE) [a]	Chemical Structure	Predicted Binding Mode in GES-5
**8**	0.89 (0.27)	0.35 (0.30)	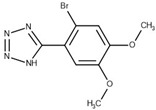	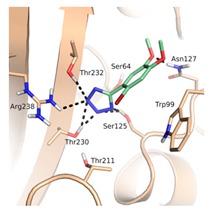
**12**	1.01 (0.22)	1.38 (0.21)	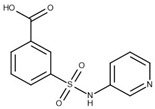	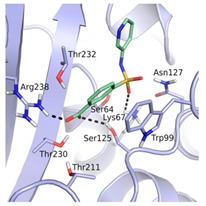
**13**	0.81 (0.29)	0.28 (0.33)	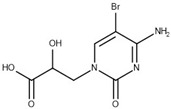	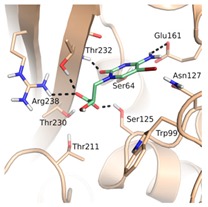
**43**	0.66 (0.32)	0.45 (0.33)	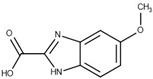	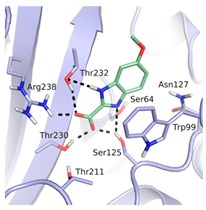

[a]LE [kcal/mol/non-hydrogen atom].

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
