# Peer review of "Targeting the Class A Carbapenemase GES-5 via Virtual Screening"

_biomolecules, 2020, doi:10.3390/biom10020304_

Round 1

Reviewer 1 Report

The manuscript of Klein and coworkers (biomolecules-707802) reports an investigation on carbapenemase inhibitors. After a screening performed ‘in silico’ against GES-5, the best compounds were also tested ‘in vitro’ to assess the inhibitor effect towards both GES-5 and KPC-2. The results obtained are not really impressive, as the inhibitory effects reported are relatively weak. On the other hand, the work seems in general to be nicely performed, and it might be of interest for a following hit-to-lead optimization of the compounds identified.

The main issues of this manuscript is that the description of the force field (or scoring function) used in the virtual screening is quite concerning. I am also reporting a number of minor issues, and in particular one regarding Figure 4.

Main points:

- Line 118-120: “Polar hydrogen atoms were added to the receptor, their positions were energy minimized and partial charges based on the AMBER force field parameters were assigned”. This protocol is quite strange for several reasons. (1) It is not clear what kind of energy minimization was performed, and in particular which force field was used (the force field is mentioned only later in the sentence, when the minimization was already performed), and whether and how the non-hydrogen atoms were fixed during the minimization. (2) AMBER is an all-atom force field, and it cannot be used when only polar hydrogens are present. (3) It is not clear what is the purpose of assigning these charges (unless an AMBER-based scoring function is later used in the docking - but this does not seem to be the case).

- Line 128-130: “As a result, the partial charges of the side chain atoms of Thr232 were adjusted to better model what was believed to be an important hydrogen bond interaction with this residue”. Modifying the partial charges is a very delicate issue, as it completely alters the force field or scoring function used, and “adjusted to better model an hydrogen bond” is totally insufficient as an explanation. How were the partial charges modified (e.g., to reproduce the equilibrium distance of the hydrogen bond, or to improve its directionality, or to mimic some other physical quantity)? Which were the partial charges used, exactly (this is important for the reproducibility of the work)? How can one trust that the impact of these modifications on the other interactions (taking place in the same binding sites during the same docking scoring) is negligible?  

Minor point:

- I have several observations about Figure 4. (1) The labeling of panel “a” is very misleading. For instance, does the single bar in between compound 14 and 17 refer to compound 15, or is it 16? (2) Panel “a” includes 37 compounds, and panel “b” includes 26 compounds. Where are the common features of those two groups of well-defined compounds described in the text of the manuscript, exactly? (3) Most of the information contained in panel “a” is obviously repeated in panel “b”. Is it possible to delete altogether panel “a”, and add to panel “b” a few more bars (each accompanied by a gap indicating a “missing” red bar, and/or by an asterisk that explains in the Caption that values for KPC-2 is not tested)?

Other minor observations:

- Line 20: “in-silico”. It is not necessary to use the hyphen, especially because two lines later it is not used for “in vitro”.

- Line 60: “@-lactams”. The symbol should be corrected.

- Line 64: “respond often only to”. Better is “often respond only to”.

- Line 66: “now available making it”. Using a comma would be better.

- Figure 1, caption: “is addressed by the carboxylate group”. The verb “addressed” is not clear.

- Line 80: “ligand efficiencies (EL)”. It should be “(LE)”.

- Line 83: “Forty-four mostly fragment-like compounds”. Better is “Forty-four compounds, mostly fragment-like”.

- Line 87: “Two of them had LEs with 029 and 0.30, resp against GES-5”. This sentence should be completely revised.

- Line 91-98: It would be better to use in the whole Section a more appropriate verb tense: “was gel filtered”, “was performed”, “was refined”, etc.

- Line 94:  “two molecules per a.s.u. and has been refined to Rfree/Rwork values”. What is “a.s.u.”? What is “Rfree/Rwork”?

- Line 96: “extra density”. It should be used “electron density” or some other specification.

- Line 117: ‘Protein Builder”. The quotation marks used are different.

- Line 123: “van-der-Waals potential”. Do not use hyphens.

- Line 166: “inhouse structure”. “In-house” is used in all cases elsewhere.

- Line 167 and 170: “rmsd”. Although the meaning may seem obvious, the abbreviation should be defined. It should also be specified that those are “positional” and “atomic” RMSD, when necessary.

- Line 169: “backbone NHs”. Another expression (e.g., “backbone NH groups”) would be better.

- Line 179: “h-bond interaction”. “Hydrogen bond” is used in all cases elsewhere.

- Line 188: “In the one setup”. Use either “in one setup” or “in the first setup”.

- Line 213-216: it would be better to stress again somewhere that these comments are referring only to the four compounds above mentioned.

- Table 1: the meaning of the values reported in parentheses should be explicitly stated in the Caption. A space should be also added in the column label "KPC-2(LE)".

- Line 234 and 241: “are very similar close to” and “differ more distant to”. These expressions are unclear. Using “similar/different in the region close/distant...” would be much better.

- Line 246: “LE of 0.26 0.22 kcal/mol/non-hydrogen atom”. This mistake must be corrected. More generally, the whole sentence it belongs to is very long and convoluted. I recommend to split it in at least two distinct sentences, and if possible also to rephrase them.

- Line 254: “contributing to the low affinity”. This sentence is ambiguous: low binding energy means high affinity. The usual way to avoid such ambiguity is using “favorable/unfavorable” instead of “low/high”.

- Line 255: “growing the compounds”. The concept is clear but the expression is terrible. A different expression should be used instead of “growing”.

- Author contribution: “Funding acquisition, Donatella Tondi”, “Project administration, Donatella Tondi”. Although nowadays funding acquisition and management are a vital part of any project, these two lines give me a very bad impression of the role of Donatella Tondi as a researcher. I am not arguing that these lines should be necessarily deleted, but please consider that my impression might perhaps be shared also by others. The section “Funding” should be already sufficient, or maybe more details (if strictly needed) could be added there.  

- Supplementary material: either explain the meaning of “n.d.” or substitute it with "-".

Author Response

We thank the reviewer for very thorough and detailed feedback. Below we answer point by point to the raised issues. 

Main points:

 - Line 118-120: “Polar hydrogen atoms were added to the receptor, their positions were energy minimized and partial charges based on the AMBER force field parameters were assigned”. This protocol is quite strange for several reasons. (1) It is not clear what kind of energy minimization was performed, and in particular which force field was used (the force field is mentioned only later in the sentence, when the minimization was already performed), and whether and how the non-hydrogen atoms were fixed during the minimization. (2) AMBER is an all-atom force field, and it cannot be used when only polar hydrogens are present. (3) It is not clear what is the purpose of assigning these charges (unless an AMBER-based scoring function is later used in the docking - but this does not seem to be the case).

Reply: This section was indeed not accurate. The ‘protonate 3D’ tool from MOE is based on the MMFF94 force field and not AMBER. We have inserted a reference to this tool so that the interested reader can find more details. The non-hydrogen atoms were fixed during minimization and we have added this detail to the text. The charges are assigned by the MOE tool but not used later, therefore we have deleted the part about assigning the partial charges.

- Line 128-130: “As a result, the partial charges of the side chain atoms of Thr232 were adjusted to better model what was believed to be an important hydrogen bond interaction with this residue”. Modifying the partial charges is a very delicate issue, as it completely alters the force field or scoring function used, and “adjusted to better model an hydrogen bond” is totally insufficient as an explanation. How were the partial charges modified (e.g., to reproduce the equilibrium distance of the hydrogen bond, or to improve its directionality, or to mimic some other physical quantity)? Which were the partial charges used, exactly (this is important for the reproducibility of the work)? How can one trust that the impact of these modifications on the other interactions (taking place in the same binding sites during the same docking scoring) is negligible?  

Reply:  The magnitude of some partial charges of side chain atoms of Thr232 were altered without changing the net residue charge. This was done in order to be able to reproduce the binding modes of ligands of related beta-lactamases. From a physical point of view, changing partial charges without changing the net charge of a residue, is a very crude way of mimicking a polarizable force field in which the partial charge of an atom is environment dependent.

Changing the partial charges of Thre232 will indeed have an effect on the total interactions in the binding site, but this is an intended consequence as this results in the correct binding mode being predicted. This approach is actually quite common when using Dock 3.x for virtual screening, see for example also Lyu, J.; Wang, S.; Balius, T. E.; Singh, I.; Levit, A.; Moroz, Y. S.; O’Meara, M. J.; Che, T.; Algaa, E.; Tolmachova, K.; et al. Ultra-Large Library Docking for Discovering New Chemotypes. Nature 2019.

We have added some more details to the text to explain why the charges were altered and by how much (lines 127-132).

Minor point:

 - I have several observations about Figure 4. (1) The labeling of panel “a” is very misleading. For instance, does the single bar in between compound 14 and 17 refer to compound 15, or is it 16? (2) Panel “a” includes 37 compounds, and panel “b” includes 26 compounds. Where are the common features of those two groups of well-defined compounds described in the text of the manuscript, exactly? (3) Most of the information contained in panel “a” is obviously repeated in panel “b”. Is it possible to delete altogether panel “a”, and add to panel “b” a few more bars (each accompanied by a gap indicating a “missing” red bar, and/or by an asterisk that explains in the Caption that values for KPC-2 is not tested)?

Reply: Figures 4a and b were adjusted to clearly label the bars.

We have added a sentence explaining how the subset of compounds shown in Fig. 4b were selected. “This subset included all compounds that showed at least 28 % inhibition at 1 mM for GES-5 and less potent compounds that appeared to be very soluble under the assay conditions.” (lines 210-212)

We prefer to keep figures a and b separate as we think this makes the results easier to read.

Other minor observations:

 - Line 20: “in-silico”. It is not necessary to use the hyphen, especially because two lines later it is not used for “in vitro”.

Reply: corrected

- Line 60: “@-lactams”. The symbol should be corrected.

Reply: corrected

- Line 64: “respond often only to”. Better is “often respond only to”.

Reply: corrected

 - Line 66: “now available making it”. Using a comma would be better.

Reply: corrected

- Figure 1, caption: “is addressed by the carboxylate group”. The verb “addressed” is not clear.

Reply: We have changed this sentence to “The carboxylate group of imipenem is oriented in the oxyanion hole formed by Ser125, Thr211, Thr230 and Thr232.”

- Line 80: “ligand efficiencies (EL)”. It should be “(LE)”.

Reply: corrected

- Line 83: “Forty-four mostly fragment-like compounds”. Better is “Forty-four compounds, mostly fragment-like”.

Reply: corrected

- Line 87: “Two of them had LEs with 029 and 0.30, resp against GES-5”. This sentence should be completely revised.

Reply: corrected. The new sentence reads as: “Two of them had LEs > 029 kcal/mol/non-hydrogen atom against GES-5.”

- Line 91-98: It would be better to use in the whole Section a more appropriate verb tense: “was gel filtered”, “was performed”, “was refined”, etc.

Reply: corrected

- Line 94:  “two molecules per a.s.u. and has been refined to Rfree/Rwork values”. What is “a.s.u.”? What is “Rfree/Rwork”?

Reply: A.s.u. stand for Asymmetric Unit. We have added this to the text.

Rfree and Rwork values are a measure of the quality of the atomic model obtained from the crystallographic data. These are very common terms in the field of X-ray crystallography and we therefore have chosen to add no further explanation to the text.

- Line 96: “extra density”. It should be used “electron density” or some other specification.

Reply: We have changed this sentence to “In the active site, an electron density blob close to Ser125 and Thr230 was interpreted as dimethyl sulfoxide ...”

- Line 117: ‘Protein Builder”. The quotation marks used are different.

Reply: corrected

- Line 123: “van-der-Waals potential”. Do not use hyphens.

Reply: corrected

- Line 166: “inhouse structure”. “In-house” is used in all cases elsewhere.

Reply: corrected

- Line 167 and 170: “rmsd”. Although the meaning may seem obvious, the abbreviation should be defined. It should also be specified that those are “positional” and “atomic” RMSD, when necessary.

Reply: corrected

- Line 169: “backbone NHs”. Another expression (e.g., “backbone NH groups”) would be better.

Reply: corrected

 - Line 179: “h-bond interaction”. “Hydrogen bond” is used in all cases elsewhere.

Reply: corrected

- Line 188: “In the one setup”. Use either “in one setup” or “in the first setup”.

 Reply: We have changed this sentence to “Therefore, two different structures were used for docking, one containing the rotamer found in GES-5 and one containing the rotamer found in KPC-2.”

- Line 213-216: it would be better to stress again somewhere that these comments are referring only to the four compounds above mentioned.

Reply: We mention in the first sentence that Ki values have been determined for a certain set of compounds and in the following sentence we comment on these values. As only for these compounds Ki values have been determined in this study, we think that this paragraph is sufficiently clear and prefer not to change the wording.

- Table 1: the meaning of the values reported in parentheses should be explicitly stated in the Caption. A space should be also added in the column label "KPC-2(LE)".

Reply: corrected

 - Line 234 and 241: “are very similar close to” and “differ more distant to”. These expressions are unclear. Using “similar/different in the region close/distant...” would be much better.

 Reply: We have changed these sentences to:

“The binding sites of GES-5 and KPC-2 are very similar, especially at the level of the here targeted oxyanion hole.”

and

“However, moving away from the oxyanion hole, the two binding sites become more different.”

- Line 246: “LE of 0.26 0.22 kcal/mol/non-hydrogen atom”. This mistake must be corrected. More generally, the whole sentence it belongs to is very long and convoluted. I recommend to split it in at least two distinct sentences, and if possible also to rephrase them.

Reply: corrected. This section now reads: “In the previous study, the top hit in terms of affinity had a Ki of 0.16 mM and a LE of 0.22 kcal/mol/non-hydrogen atom and the top hit in terms of LE had a Ki of 0.50 mM and a LE of 0.26 kcal/mol/non-hydrogen atom [16]. Here, the top hit has a Ki of 0.66 mM and a LE of 0.32 kcal/mol/non-hydrogen atom.” (lines 248-251)

- Line 254: “contributing to the low affinity”. This sentence is ambiguous: low binding energy means high affinity. The usual way to avoid such ambiguity is using “favorable/unfavorable” instead of “low/high”.

Reply: We have changed the sentence to “.. further contributing to the unfavorable potency ..”.

- Line 255: “growing the compounds”. The concept is clear but the expression is terrible. A different expression should be used instead of “growing”.

Reply: Fragment growing is a standard term in the field of fragment-based drug design. Therefore, we prefer to keep this expression in the text.

- Author contribution: “Funding acquisition, Donatella Tondi”, “Project administration, Donatella Tondi”. Although nowadays funding acquisition and management are a vital part of any project, these two lines give me a very bad impression of the role of Donatella Tondi as a researcher. I am not arguing that these lines should be necessarily deleted, but please consider that my impression might perhaps be shared also by others. The section “Funding” should be already sufficient, or maybe more details (if strictly needed) could be added there.  

Reply: We have simply filled out this section according to the journal guidelines and therefore prefer to not change it except if instructed to do so by the journal editors.

- Supplementary material: either explain the meaning of “n.d.” or substitute it with "-".

Reply: An explanation has been added.

Reviewer 2 Report

Title: Targeting the class A carbapenemase GES-5 via virtual screening

This manuscript describes virtual screening and experimental validation of inhibitors of GES-5. This class of carbapenemase is important to broad spectrum antibiotic resistance, making this work of biomedical significance. Overall, the manuscript is well written and the methodology appears sound, although I should stress that I am far from an expert in virtual screening. What is not as clear is how this work differs from the authors’ previous work (reference 16) describing a very similar process for the same enzyme. While the structures of the compounds identified in each screen are significantly different, they are all of similarly low potency. The authors should use the Discussion to expand on the differences between the two studies and the additional insights gained from the current work to validate its publication. There are a few other revisions as indicated below:

Line 34: “thread” should be “threat” Line 60: The beta symbol is for beta-lactams needs to be corrected. In this same area, I do not understand what “higher and more hydrolytic activity” means. I suggest deleting “and more” Lines 70-71: References for the 5 crystal structures of GES-5 should be included Section 2.1, Fig. 1: It is not clear how the imipenem shown in Fig. 1 got into the structure. Was it soaked into the crystals or co-crystallized? It is especially confusing that the experimental section indicates that density for DMSO was found in the active site that would seem to clash with this ligand. Line 100: “analogue” should be “analogous” Line 166: Table S1 should be referred to here 4: In 4a only 37 bars are present in the figure for 44 compounds. It should be clarified that those compounds not tested were due to insolubility and the compound number for each bar needs to be indicated. How were the compounds selected to test against KPC-2 in Fig 4b?

Author Response

We thank the reviewer for positive and constructive feedback. Below, we answer point by point to the raised issues.

What is not as clear is how this work differs from the authors’ previous work (reference 16) describing a very similar process for the same enzyme. While the structures of the compounds identified in each screen are significantly different, they are all of similarly low potency. The authors should use the Discussion to expand on the differences between the two studies and the additional insights gained from the current work to validate its publication.

Reply: As suggested, we have elaborated on the differences between both studies and modified the discussion (lines 247-256).

Line 34: “thread” should be “threat”

Reply: corrected

Line 60: The beta symbol is for beta-lactams needs to be corrected. In this same area, I do not understand what “higher and more hydrolytic activity” means. I suggest deleting “and more”

Reply: corrected

Lines 70-71: References for the 5 crystal structures of GES-5 should be included

Reply: corrected. In addition, we have added references to other GES crystal structures.

Section 2.1,

Fig. 1: It is not clear how the imipenem shown in Fig. 1 got into the structure. Was it soaked into the crystals or co-crystallized? It is especially confusing that the experimental section indicates that density for DMSO was found in the active site that would seem to clash with this ligand.

Reply: The structure of the imipenem complex was determined previously by another group (Ref 12). The apo-structure of GES-5 disclosed in our paper does not contain an inhibitor. Hence, there is space for a DMSO molecule in the active site.

Line 100: “analogue” should be “analogous”

Reply: corrected

Line 166: Table S1 should be referred to here

Reply: done

Figure 4: In 4a only 37 bars are present in the figure for 44 compounds. It should be clarified that those compounds not tested were due to insolubility and the compound number for each bar needs to be indicated. How were the compounds selected to test against KPC-2 in Fig 4b?

Reply: We have added a statement that 7 compounds were not soluble under assay conditions to the text (lines 204-207). Further, figure 4a was changed to include a compound number for each bar.

We have added a sentence to explain which compounds were tested against KPC-2: “This subset included all compounds that showed at least 28 % inhibition at 1 mM for GES-5 and less potent compounds that appeared to be very soluble under the assay conditions.” (lines 210-212)

Reviewer 3 Report

The paper entitled “Targeting the class A carbapenemase GES-5 via virtual screenings” describes an in-silico screening against the GES-5 carbapenemase, using a lead-like compound library of commercially available compounds. The most promising candidates were selected for in vitro validation in biochemical assays against recombinant GES-5.

Specific comments:

- At line 80, ligand efficiency was shortened as EL instead LE, while the authors used LE later in the text. Please check.

- Reference at line 193 should be incorrect, please check.

- Under the result section, the inhibition percentages of the 44 compounds, tested at 1 mM concentration in the assay against GES-5, are shown in figure 4a and in table S2, as reported at line 208. However, in table S2 some inhibition values are “n.d” or “not tested for solubility limit” (for example, compounds 16 and 11). Why do they appear in figure 4a? The authors should explain this inconsistency.

Furthermore, at line 208-209 the authors reported: “A subset of the compounds was also tested against KPC-2 to evaluate their ability to inhibit other class A carbapenemases (Fig. 4b)”, but in figure 4b inhibition percentages are shown for all 44 compounds.

- At line 215-216, the authors stated: “Except for compound 12, KPC-2 was more strongly inhibited by these compounds than GES-5, with Ki values ranging from 0.28 μM to 1.38 mM.” However, in table 1 all Ki values are expressed in mM. Please check.

- Reference 1 (line 186) should be better specified to let the reader find it.

Author Response

We thank the reviewer for positive and constructive feedback. Below, we address the raised issues point by point.

- At line 80, ligand efficiency was shortened as EL instead LE, while the authors used LE later in the text. Please check.

Reply: corrected

- Reference at line 193 should be incorrect, please check.

Reply: corrected

- Under the result section, the inhibition percentages of the 44 compounds, tested at 1 mM concentration in the assay against GES-5, are shown in figure 4a and in table S2, as reported at line 208. However, in table S2 some inhibition values are “n.d” or “not tested for solubility limit” (for example, compounds 16 and 11). Why do they appear in figure 4a? The authors should explain this inconsistency.

Reply: Not tested compounds were not included in figure 4a, but some compound labels were lacking in this figure which made it impossible to read the figure correctly. This has been corrected.

Furthermore, at line 208-209 the authors reported: “A subset of the compounds was also tested against KPC-2 to evaluate their ability to inhibit other class A carbapenemases (Fig. 4b)”, but in figure 4b inhibition percentages are shown for all 44 compounds.

Reply: This misunderstanding is again due to the missing labels. A corrected figure has been inserted.

- At line 215-216, the authors stated: “Except for compound 12, KPC-2 was more strongly inhibited by these compounds than GES-5, with Ki values ranging from 0.28 μM to 1.38 mM.” However, in table 1 all Ki values are expressed in mM. Please check.

Reply: µM should read mM. This has been corrected.

- Reference 1 (line 186) should be better specified to let the reader find it.

Reply: corrected

Round 2

Reviewer 1 Report

I am very satisfied about the modifications in the revised version, and I am happy to support publication of this work in Biomolecules. I would like to point out only two minor nitpicks that could be corrected even at proof stage:

Line 87: "> 029 kcal/mol/non-hydrogen atom". It seems a "." is missing.

Line 118-119: "polar hydrogen atoms were added to the receptor and their positions were energy minimized while keeping the protein atoms rigid[22]". It still sounds quite strange that a minimization (extensive, or even minor) is necessary, so it might be considered changing this to a simpler "polar hydrogen atoms were added to the receptor while keeping the protein atoms rigid[22]".

Author Response

We thank the reviewer for quickly reviewing the revisions.
Below we have answered to the remaining two issues.

Line 87: "> 029 kcal/mol/non-hydrogen atom". It seems a "." is missing.
Corrected

Line 118-119: "polar hydrogen atoms were added to the receptor and their positions were energy minimized while keeping the protein atoms rigid[22]". It still sounds quite strange that a minimization (extensive, or even minor) is necessary, so it might be considered changing this to a simpler "polar hydrogen atoms were added to the receptor while keeping the protein atoms rigid[22]".

For some polar groups, there are several possible orientations for hydrogen atoms which are allowed from a purely geometrical point of few. The most likely hydrogen atom is depending on the formed hydrogen-bond network with neighboring groups. Therefore, we have chosen to minimize the positions of hydrogen atoms to take this into account.

Reviewer 2 Report

The authors have responded to nearly all of my concerns and the revised manuscript is improved. However, I find it inappropriate that the legend for Fig. 1 indicates that the displayed structure is PDB code 6TS9 when that structure does not contain a bound imipenem. I suggest that the authors simply display the structure from reference 12 and cite the correct pdb code. Alternatively, they can explain that imipenem has been docked into the active site of structure 6TS9. 

Author Response

We thank the reviewer for quickly reviewing the revisions.
Below is our reply to the remaining issue.

I find it inappropriate that the legend for Fig. 1 indicates that the displayed structure is PDB code 6TS9 when that structure does not contain a bound imipenem. I suggest that the authors simply display the structure from reference 12 and cite the correct pdb code. Alternatively, they can explain that imipenem has been docked into the active site of structure 6TS9.

The structure in Fig. 1 is indeed the covalent complex. However, we have listed the wrong PDB code. It should be 4H8R, this has been corrected in the manuscript.